# Identification of Bitter Peptides in *Lilium lancifolium* Thunb.; Peptidomics, Computational Simulation and Cellular Functional Assays

**DOI:** 10.3390/foods14234056

**Published:** 2025-11-26

**Authors:** Zhuang Dong, Xiaohong Zhong, Mengshan Sun, Peng Huang, Yuedong He, Haiyuan Gong, Li Zhou, Jianguo Zeng, Wei Xiang

**Affiliations:** 1College of Horticulture, Hunan Agricultural University, Changsha 410128, China; dzlebron0701@163.com (Z.D.);; 2Hunan Key Laboratory of Traditional Chinese Veterinary Medicine, Hunan Agricultural University, Changsha 410128, China; 3Yuelushan Laboratory, Changsha 410128, China; 4Hunan Institute of Nuclear Agriculture Sciences and Chinese Herbal Medicines, Changsha 410125, China; 5College of Veterinary Medicine, Hunan Agricultural University, Changsha 410128, China; 6College of Bioscience and Biotechnology, Hunan Agricultural University, Changsha 410128, China

**Keywords:** *Lilium lancifolium* Thunb., peptidomics, bitter peptides, TAS2R14 receptor, molecular docking and dynamics simulation, signaling pathway

## Abstract

*Lilium lancifolium* Thunb., as a predominant variety of medicinal and edible lilies, has long been renowned in traditional medicine for “moistening the lungs, relieving coughs, and calming the mind to soothe the heart.” The bitter taste formation in *L. lancifolium* is predominantly attributed to secondary metabolites such as alkaloids, this study explores an alternative mechanism underlying taste divergence among *Lilium brownii* var. *viridulum*, and *Lilium pumilum* DC, proposing a foundational scientific question: Are peptides one of the important sources of bitterness in *Lilium lancifolium* Thunb.? Peptidomic analysis identified 8479 peptide sequences, with 46.27% upregulated in *L. lancifolium* flesh. Through high-throughput molecular docking with the bitter taste receptor TAS2R14, 214 candidate bitter peptides were identified, showing the strongest average binding affinity (−119.73 kcal/mol). Molecular dynamics simulations further demonstrated that four of these peptides formed stable interactions with key residues in TAS2R14. Cellular assays confirmed TAS2R14 activation by these peptides, as indicated by enhanced EGFP reporter fluorescence, upregulation of downstream signaling molecules (*GNAT1*, *PLCB2*, *TRPM5*), decreased cAMP levels, and increased IP_3_ accumulation. Transcriptomic analysis further indicated that bitter peptides mediate taste transduction primarily through neuroactive receptor interaction pathways. These findings represent the first identification of bitter peptides as a key source of bitterness in *L. lancifolium* and elucidates their transduction mechanism combining peptidomics, computational simulation, and cellular validation. Our study provides a methodological framework for exploring flavor substances in other plant-derived foods.

## 1. Introduction

The genus *Lilium* comprises approximately 110 species with global distribution. Certain taxa in this genus, particularly *Lilium lancifolium* Thunb., produce bulbs used in traditional medicinal and culinary practices throughout East Asia, specifically in China, Japan and Korea [1,2]. Current research indicates that lily bulbs primarily contain steroidal saponins, phenylpropanoid derivatives and polysaccharides [3,4,5,6], and exhibit bioactivities such as lung-moistening, cough suppression, immunomodulation, along with antioxidant, hypoglycemic and anti-inflammatory effects [7,8]. Classical texts of Traditional Chinese Medicine, such as the *Zhong Yao Da Ci Dian* and *Zhong Hua Ben Cao, document* the use of fresh lily bulbs in treating symptoms including “lung-heat cough, dry and persistent cough,” as well as “lung stagnation and heat, accompanied by restlessness and cough [9,10]. The Chinese Pharmacopoeia documents *L. lancifolium*, *L. brownii* var. *viridulum*, and *L. pumilum* as source species for medicinal lily bulbs and the former two comprising more than 98% of commercial distribution [11].

However, this taste divergence stems from the specificity of chemical composition yet related research on lily bulbs remains insufficient. During processing or digestion, many plant-derived proteins generate bitter-tasting peptides [12,13], yet the protein and peptide composition of *L. lancifolium* has received relatively little research attention. This study seeks to determine whether polypeptide components constitute a source of bitterness in *L. lancifolium*, thereby establishing a theoretical foundation for accurate bitterness assessment and systematic regulation of its edible and medicinal quality. Conventionally, bitterness in plants is predominantly attributed to secondary metabolites such as alkaloids compounds that have garnered extensive research attention due to their intense bitterness [12,13]. Food bitterness arises primarily through synergistic interactions among multiple constituents where peptides function as principal mediators [14]. Bitter peptides derived from soybean protein hydrolysis constitute the dominant source of bitterness in fermented soybean foods [15]. However, the potential role of peptides in bitter taste formation in lilies, especially *L. lancifoli* remain unclear.

The molecular basis of bitter taste perception involves 25 human bitter taste receptor subtypes encoded by the human genome and expressed in type II taste buds. Despite low sequence homology (<20%) among TAS2Rs members, all retain the characteristic 7-transmembrane domain and highly variable extracellular loops. This structural variability confers the ability to recognize diverse bitter compounds [16,17]. Upon activation, TAS2Rs initiate signal transduction through their cognate G-protein through dual pathways: the Gβγ subunit activates phospholipase Cβ2 (*PLCB2*), catalyzing inositol trisphosphate (IP_3_) generation and Ca^2+^ release [18] while the *GNAT1* subunit activates phosphodiesterase to reduce intracellular cyclic cAMP levels. Elevated intracellular Ca^2+^ activates the transient receptor potential channel *TRPM5*. This activation induces taste cell depolarization [19], resulting in the transmission of neural signals to the central nervous system. Advances in bioinformatics and computational chemistry have established molecular docking and dynamics (MD) simulation as indispensable techniques for identifying bitter compounds.

In previous research, computational simulations and molecular docking identified 36 and 14 potential bitter peptides from wheat protein hydrolysates [20] and fermented bean curd [21], respectively. Also, computational analyses have demonstrated that Jinhua ham-derived bitter peptides (PKAPPAK, VTDTTR, YIIEK) activate TAS2R16 through hydrophobic interactions and hydrogen bonding [22]. Parallel mechanisms were reported for ADM/ADW peptides from rainbow trout nebulin hydrolysate, which bind critical residues (Thr86, Asp168, Phe247) of TAS2R14 to inhibit receptor activation and mitigate bitterness perception [23]. Given the increasing market demand for lilies, systematic analysis of bitter components is essential for targeted quality improvement and novel functional food/pharmaceutical development. This study aims to identify bitter peptides in *L. lancifolium* and integrates peptidomics to compare the peptide profiles in bitter *L. lancifolium*, low-bitter *L. brownii* var. *viridulum* and sweet *L. pumilum*. MD simulations validated the key binding residues. A HEK293T cell line that stably expresses TAS2R14-EGFP was established to simultaneously monitor peptide-induced fluorescence dynamics, mRNA expression of downstream effectors (*PLCB2*, *TRPM5*, *GNAT1*) and intracellular second messengers. Integrated transcriptomics identified differentially expressed genes with functional enrichment that elucidates the bitter peptide signaling network. This study pioneered the systematic identification of bitter peptides in *L. lancifolium* and validated four bitter peptides, while confirming their TAS2R14-mediated mechanisms. This establishes a methodological framework for analyzing bitter compounds in plants.

## 2. Materials and Methods

### 2.1. Materials

Mature and disease-free bulbs of three lily varieties (*L. lancifolium*, *L. brownii* var. *viridulum*, and *L. davidii* var. *willmottiae*) were rinsed with purified water, flash-frozen in liquid nitrogen, and stored at −80 °C for 24–48 h. Following thawing, rapid dissection of epidermis and flesh scale tissues generated six samples: LLT_E (epidermis of *L. lancifolium*), LLT_F (flesh of *L. lancifolium*), LBVV_E (epidermis of *L. brownii* var. *viridulum*), LBVV_F (flesh of *L. brownii* var. *viridulum*), LDVW_E (epidermis of *L. davidii* var. *willmottiae*), and LDVW_F (flesh of *L. davidii* var. *willmottiae*). The samples were lyophilized followed by cryogenic pulverization using a ball mill apparatus from Changsha Miqi Instrument Equipment Co., Ltd. (Changsha, China). The processed samples were then cryopreserved at −80 °C under nitrogen atmosphere pending analysis. Three biological replicates per tissue type generated a total of 18 samples.

### 2.2. Methods

#### 2.2.1. Electronic Tongue Analysis and Sensory Evaluation

A human sensory panel was established to evaluate the bitterness intensity of the synthesized peptides. Eight panelists (aged 18–45 years, including students and staff) were recruited from the laboratory and surrounding communities. All participants were confirmed to be non-smokers, with no known taste disorders or food allergies, and were not taking medication that could affect taste perception. Prior to formal evaluation, all panelists underwent a standardized training session. This session included: An introduction to the sensory evaluation procedure and the specific task of rating bitterness intensity; Familiarization with the reference standards: a 0.05% (*w*/*v*) caffeine solution was used as an anchor for “moderate bitterness” and deionized water was used as a “no bitterness” control; Practice evaluations of the reference samples and a series of peptide solutions at different concentrations to ensure consistent understanding and use of the rating scale. During the formal testing, each peptide sample and control was evaluated in triplicate by each panelist on separate days to assess reproducibility. The presentation order of samples was randomized for each session to avoid bias. Panelists were instructed to rinse their mouths thoroughly with deionized water between samples and to wait for a minimum of 1 min interval to minimize carry-over effects. The evaluation was conducted in a room with a temperature of 25 °C, an odor-free environment, and a light intensity of 4000 lux. Before each tasting test, evaluators were required to rinse their mouths with 50 mL of purified water for 15 s, and this procedure was repeated three times. Detailed evaluation criteria are provided in Appendix A. The research involving human participants in this study was conducted in accordance with the ethical standards of the Declaration of Helsinki. The protocol received ethical approval from the Biomedical Research Ethics Committee of Hunan Agricultural University (Approval No: 2025-194). Informed consent was obtained from all individual participants involved in the study. Informed consent was obtained from all individual participants involved in the study.

An electronic tongue system (SA402B, INSENT Co., Atsugi-chi, Japan) was employed with the specific aim of evaluating and analyzing the bitter taste profile. The sensor arrays were processed using Principal Component Analysis (PCA) for initial pattern recognition and sample differentiation. Furthermore, the quantitative analysis of the key taste traits (such as bitterness intensity) was achieved by building a prediction model based on Artificial Neural Network (ANN) algorithms. All the samples were subjected to a 10-fold dilution, followed by accurate dispensing of 50 mL of aliquots in three measurement cups for analysis. The pH and conductivity of the solution were measured to confirm stability before the electronic tongue analysis. A reference solution containing 3 mM KCl and 30 mM tartaric acid was prepared. Sensor cleaning solutions consisted of two distinct formulations: the anode solution containing 0.01 mM HCl, ethanol, and ultrapure water; and the cathode solution comprising saturated KCl, 0.01 M KOH, ethanol, and ultrapure water [24]. Sensors were subjected to baseline calibration in a reference solution prior to analytical measurements. Taste analysis consisted of four consecutive 30 s sample exposure intervals, each followed by a 3 s rinse cycle in reference solution. Quadruplicate measurements per sample were recorded, and analytical data was derived from the mean values of the final three measurement cycles.

#### 2.2.2. Peptide Extraction

Aliquots (20 mg) of the sample were homogenized in 1.5 mL of ice-cold RIPA lysis buffer containing 1 mM phenylmethylsulfonyl fluoride and 2 mM ethylenediaminetetraacetic acid [25]. After 5 min pulse sonication on ice and centrifugation at 15,000× *g* for 10 min at 4 °C, the supernatant underwent sequential reduction with 10 mM dithiothreitol at 56 °C for 30 min and alkylation using 20 mM iodoacetamide under light-protected conditions for 30 min. Protein precipitation was achieved by 20% *w*/*v* trichloroacetic acid incubation at 4 °C for 2 h, followed by centrifugation at 12,000× *g* for 10 min at 4 °C and two ice-cold acetone washes [26]. Chloroform extraction removed residual lipids. Peptides were purified through C_18_ solid-phase extraction columns, vacuum-dried, and quantified via bicinchoninic acid assay [27].

#### 2.2.3. Lc-Ms/Ms Analysis and Data Processing

The peptides (200 ng) were separated using a NanoElute UHPLC system (Bruker, Billerica, MA, USA) using a reverse-phase C_18_ column (IonOpticks, Melbourne, Australia; 25 cm × 75 μm, 1.6 μm particle size) maintained at 50 °C with a flow rate of 300 nL/min. The mobile phases consisted of (A) 0.1% (*v*/*v*) aqueous formic acid and (B) 0.1% (*v*/*v*) formic acid in acetonitrile. The gradient profile was as follows: 0–22 min (2% → 22% B), 22–25 min (22% → 35% B), 25–27 min (35% → 80% B), followed by isocratic elution at 80% B from 27–30 min. Eluted peptides were analyzed on a tmsTOF Pro 2 (Bruker, Bremen, Germany) mass spectrometer operated in DDA-PASEF mode. Key parameters included: positive ion mode; *m*/*z* range 100–1700; ion mobility range 0.7–1.4 Vs/cm^2^; capillary voltage 1500 V; nitrogen drying gas temperature 180 °C at 3 L/min; PASEF settings: 10 MS/MS scans per cycle (cycle time 1.17 s), precursor charge states 1–5 (corrected from 0–5), dynamic exclusion duration 0.4 min, intensity threshold 2500 counts; collision energy ramp 20–59 eV; quadrupole isolation widths: 2 Th for *m*/*z* < 700 and 3 Th for *m*/*z* > 800.

#### 2.2.4. Database Search and Quantification

The search parameters for the database were specified with precursor and fragment mass tolerances of ±20 ppm. The carboxymethylation of cysteine residues was set as a fixed modification, while the variable modifications included methionine oxidation and N-terminal acetylation, allowing a maximum of three modifications per peptide. Label-free quantification utilized the MaxLFQ algorithm within IonQuant with match-between-runs retention time alignment. The identification of peptides and proteins was filtered at 1% false discovery rate, excluding common contaminants (e.g., keratins, trypsin).

#### 2.2.5. Homology Modeling, Molecular Docking, and Md Simulation

The three-dimensional structures for the 25 human bitter taste receptors (TAS2Rs) under investigation were acquired as follows: (1) Experimentally resolved CryoEM (cryogenic electron microscopy) structures for TAS2R14 and TAS2R46 were retrieved from the PDB database (https://www.rcsb.org/, accessed on 19 January 2025); (2) Structures for the remaining 23 receptors were constructed using the AlphaFold3 structure prediction method following the protocol described in Shams Nosrati et al. [28]. Molecular docking of the identified bitter peptides to the TAS2Rs receptors was performed using Discovery Studio 2019 (v19.1, BIOVIA). Peptide structures were generated using ChemDraw (version 25) and exported as MOL files; receptor protein structures were preprocessed through removal of crystallographic waters, hydrogen atom placement, and protonation state assignment with corresponding charge parameterization. Semi-flexible docking of the 214 bitter peptides against all 25 TAS2Rs targets was executed via the CDOCKER module. Key docking parameters included: CHARMM force field, generation of 10 stochastic ligand conformations, and a van der Waals energy threshold of 300 kJ/mol for pose generation. Four peptide-receptor complexes exhibiting optimal mass spectrometry response values, significant fold-change, and favorable binding energy were selected for further analysis. The docked conformations were visualized and analyzed using PyMOL (version 2.04) [29]. Specific protein-ligand interaction types, distances, and frequencies were statistically analyzed using Maestro (version 11) [30]. MD simulations were performed using GROMACS (version 2023.3). The simulation workflow sequentially executed energy minimization, NVT ensemble equilibration, and NPT ensemble equilibration. Energy minimization used the steepest descent algorithm with a step size of 0.01 nm until it reached a convergence threshold of 1000 kJ·mol^−1^·nm^−1^. NVT equilibration was performed for 100 ps at 310.15 K using the V-rescale thermostat. Finally, NPT equilibration proceeded for 100 ps with pressure regulated at 1 bar using the Berendsen barostat; and a 100 ns production run. Key simulation parameters included: LINCS algorithm for constraining bonds involving hydrogen atoms; Particle Mesh Ewald method for long-range electrostatic interactions with a 1.2 nm cut off; and an integration time step of 2 fs. Binding free energies were estimated using the Molecular Mechanics/Generalized Born Surface Area (MM/GBSA) method. The trajectory analysis encompassed the calculation of the root mean square deviation (RMSD), the root mean square fluctuation (RMSF), the radius of gyration (Rg) and the dynamic hydrogen bond occupancy.

#### 2.2.6. Sensory Evaluation of Total Peptides and Four Candidate Bitter Peptides

Total peptides were extracted according to Section 2.2.2. Four candidate peptides (G AAGGSLYPNWCK, ENLPGGDQEKIH, KGTEAYLLANPDAYV, GGSPVWKLDSSEPNGQRYVT) identified in Section 2.2.3 were synthesized by Hangzhou Dangang Biotechnology (Hangzhou, China; ≥98% purity by HPLC-MS), designated Pep A to Pep-D. The mass spectrometry detection report is shown in Appendix A. Caffeine standard (purity ≥ 98%) was purchased from Shanghai Yuanye Bio-Technology Co., Ltd. (Shanghai, China). Sensory analysis was carried out following the procedure detailed in Section 2.2.1.

#### 2.2.7. Construction of Egfp-Tas2r14 Stable Cell Line and Peptide Ligand-Induced Fluorescence Imaging

A HEK293T cell line that stably expresses TAS2R14 was established using the VIRUS-Free^TM^ lentiviral system. The experimental vector pPB [Exp]-EF1A>EGFP-CAG>TAS2R14-PGK>Puro encoded the target gene. The control vector pPB [Exp]-EF1A>EGFP-CAG>[ORF-stuffer]-PGK>Puro contained a non-functional stuffer sequence. Log-phase cells cultured in Dulbecco’s Modified Eagle Medium-High glucose (DMEM-H) medium supplemented with 10% Fetal Bovine Serum (FBS) and 1% penicillin/streptomycin (37 °C, 5% CO_2_) were harvested, washed with Phosphate-Buffered Saline (PBS), and resuspended in electroporation buffer (100 μL). Cell suspensions were mixed with endotoxin-free plasmid DNA and transfected using the Neon™ electroporation system (Thermo Fisher Scientific, Waltham, MA, USA; Cat# MPK5000). Stable polyclonal populations were selected using 4.0 μg/mL puromycin, a concentration optimized through kill-curve assays, yielding both experimental and control cell pools designated as 293T#TAS2R14#Pool and 293T#control#Pool, respectively. Expanded cells were verified mycoplasma-free by Polymerase Chain Reaction (PCR)and cryopreserved in freezing medium (60% basal medium, 30% FBS, 10% Dimethyl sulfoxide (DMSO)) under liquid nitrogen vapor phase. Reverse Transcription quantitative Polymerase Chain Reaction (RT-qPCR) analysis (ΔΔCt method, GAPDH reference) confirmed 275.45-fold TAS2R14 upregulation versus control (Appendix A). Peptides A–D and total extract were dissolved in PBS to 0.1, 0.2, 0.5, and 1.0 mg/mL working concentrations. Stable cells (3 × 10^3^ cells/well) were seeded in 96-well plates, incubated overnight (37 °C, 5% CO_2_), and treated with peptides for 1 h. Ligand-induced fluorescence was captured using a confocal microscope (40× objective, Green Fluorescent Protein (GFP) filter set).

#### 2.2.8. Detection of Signaling Molecules Following Peptides Stimulation

RNA was extracted from cell pellets containing 1 × 10^6^ cells with TRIzol^TM^ reagent. After 20 s of vortex mixing, the samples were incubated on ice for 5 min and centrifuged at 12,000× *g* for 10 min at 4 °C. The resulting supernatant was combined with 200 μL of chloroform, incubated for 2 min, and centrifuged at 12,000× *g* for 10 min at 4 °C. RNA was precipitated from the aqueous phase by adding an equal volume of isopropanol with 15 min of incubation, followed by centrifugation at 12,000× *g* for 15 min at 4 °C. The resulting pellet was washed two times with 75% ethanol at 7500× *g* for 5 min at 4 °C, air-dried, and finally dissolved in 40 μL nuclease-free water. cDNA synthesis was performed using 500 ng RNA with Oligo(dT)_18_ primers and PrimeScript™ II Reverse Transcriptase (Takara Bio, Kusatsu, Japan) in 20 μL reaction volumes, involving incubation at 42 °C for 60 min and enzyme inactivation at 70 °C for 15 min. RT-qPCR assays were conducted in 20 μL reaction systems containing SYBR™ FAST Master Mix (KAPA Biosystems, Wilmington, MA) and 0.25 μM gene-specific primers targeting *TAS2R14*, *PLCB2*, *GNAT1*, and *TRPM5* (sequences detailed in Appendix A). Thermal cycling parameters comprised an initial denaturation at 95 °C for 3 min; 40 cycles of denaturation at 95 °C for 5 s and annealing/extension at 60 °C for 20 s; followed by a melt curve analysis from 65 °C to 95 °C at a ramp rate of 0.5 °C per second. Relative gene expression was quantified via the 2^(−ΔΔCt) method normalized to GAPDH, with technical triplicates performed for each sample. HEK293T cells were designated peptide-treated experimental and untreated control groups. IP_3_ and cAMP concentrations were quantified using competitive ELISA kits according to manufacturer protocols: the IP_3_ assay kit (Cat# H423-1-2; Nanjing Jiancheng Bioengineering Institute, Nanjing, China) and cAMP assay kit (Cat# JL13253; Jianglai Biotech, Shanghai, China). Cell viability was assessed via CCK-8^®^ assay (Dojindo Laboratories, Kumamoto, Japan). *TAS2R14*-expressing HEK293T cells (1.0 × 10^4^/well) in Opti-DMEM + 4% FBS were seeded into 96-well plates. After 24 h incubation (37 °C, 5% CO_2_), peptides were added to final concentrations (0.1–1 mg/mL) for 1 h. CCK-8^®^ solution (10 μL) was added, incubated (2 h, 37 °C), and absorbance measured at 450 nm (reference 650 nm) using a temperature-controlled microplate reader. Viability was calculated: [(A_sample−A_blank)/(A_control−A_blank)] × 100% with three biological replicates.

#### 2.2.9. Sequencing of the Transcriptome and Validation of Differentially Expressed Genes by RT-qPCR

Total RNA was isolated from 27 samples representing nine experimental groups with three biological replicates per group. These groups included total peptides at concentrations of 0.1 to 1 mg/mL, and Pepetide C at four concentrations: 0.1, 0.2, 0.5 and 1 mg/mL, and untreated controls, using the TRIzol^TM^ method detailed in Section 2.2.8. Strand-specific RNA libraries were constructed using the NEBNext Ultra II RNA Library Prep Kit and subjected to 150 bp pair-end sequencing on an Illumina NovaSeq 6000 platform. The raw reads were processed through quality control using Fastp (version 0.23.2), aligned with the human reference genome GRCh38 (Ensembl release 104) with HISAT2 (version 2.2.1), and quantified at the gene level using featureCounts (version 2.0.3). Differential expression analysis implemented with DESeq2 version 1.38.3 revealed significant transcripts according to predefined criteria: absolute log_2_-fold change ≥ 1 combined with adjusted *p*-value < 0.05, assessed through treatment-versus-control comparisons. KEGG pathway enrichment employed clusterProfiler v4.6.2 on treatment-specific DEGs. The key DEGs (*ATF4*, *PPP2R2A*, *CCNE2*, *PPP2R5A*, *PDE7A*, *PDE6D*, *PDE8A*) were validated by RT-qPCR according to Section 2.2.6. The primers in Appendix A amplified the original RNA samples in triplicate.

## 3. Results

### 3.1. The Peptidomic Basis of Bitterness Formation in L. lancifolium and Electronic Tongue Validation

A total of 8479 peptides from the epidermis and flesh tissues of *L. davidii* var. *willmottiae*, *L. lancifolium*, and *L. brownii* var. *viridulum* were identified. Principal component analysis revealed different clustering among different groups. PC1 and PC2 explained 46.9% and 20.31% of the variance with a cumulative total of 67.21% (Figure 1A). The tight clustering of samples within the same group validated experimental reproducibility. The LLT_F group exhibited significant peptidomic divergence, showing 46.27% of the peptides being upregulated compared to other groups (Figure 1B). Comparative analysis revealed differential upregulation of peptides: LLT_E exhibited 891 upregulated peptides versus LBVV_E and 895 versus LDVW_E; LLT_F demonstrated 4480 relative to LBVV_F and 3895 relative to LDVW_F (Figure 1C–F).

WPCNA identified a turquoise module containing 1144 peptides that showed a significant correlation with LLT_F (r = 0.52, *p* = 0.03; Figure 2A). GO enrichment analysis of differential peptides in LLT_F revealed ribosomal functions, including structural components of ribosomes and translation (Figure 2B). These findings indicate enhanced translational activity that may generate bitter precursors via hydrophobic residues, particularly leucine/isoleucine-rich motifs. Further KEGG analysis enriched endoplasmic reticulum protein processing, nucleocytoplasmic transport, and MAPK signaling pathways (Figure 2C). Electronic tongue analysis of total peptides extracted from six groups demonstrated that LLT_F exhibited the highest bitterness intensity at an equivalent concentration. Based on electronic tongue analysis, the fleshy layer of LLT_F was the most bitter among the six sample groups with LBVV_F ranking second. No obvious bitterness was detected in any of the epidermal tissues (Appendix A).

### 3.2. High-Throughput Screening of Bitter Peptides in L. lancifolium and Their Receptor Interaction Networks

Analysis of four comparative groups revealed 214 upregulated peptides (MS response intensity ≥ 10) in LLT_F, accounting for 47.72% of total peptide abundance. Given that taste perception is predominantly mediated by high-abundance components [31], we prioritized these 214 peptides for high-throughput molecular docking against all 25 human bitter taste receptors using the CDOCKER module in Discovery Studio. Eight receptors demonstrated effective binding (Table 1), characterized by key interactions including hydrogen bonds, hydrophobic forces, or π-π stacking, coupled with binding energies ≤ −5 kcal/mol based on established criteria [32].

Among these receptors, TAS2R14 and TAS2R46 bound the highest peptide counts at 76 and 121 peptides, respectively. Significantly, TAS2R14 exhibited a stronger average binding energy (−119.73 kcal/mol) than TAS2R46 (−114.88 kcal/mol). Due to its superior binding energy and the well-documented role of TAS2R14 as a broad-spectrum bitter receptor [33], this receptor was prioritized for validation of *L. lancifolium*-derived bitter peptides. The peptide of the four top-performing peptides were selected based on optimal mass spectrometry response values, fold-change, and binding energy, revealed distinctive interaction profiles with TAS2R14: Peptide A engaged 34 residues via intermolecular forces, including 11 basic residues (Lys30, Arg34, Arg38, Lys40, Lys57, Lys110, Lys123, Arg125, Lys127, Lys128, Lys244) mediating cation-π interactions and 4 acid residues (Asp105, Glu241, Asp245, Asp333) Peptide Hydrogen bonding involved 13 residues (Arg34, Arg38, Lys110, Lys30, Lys127, Glu241, etc.) at 2.9–3.2 Å (Figure 3A,E), where the basic residues (Arg/Lys) served as donors and the acid residues (Glu) as acceptors. Peptide B exhibited broader interactions across 36 residues, featuring anion-π forces centered on Asp105, Glu241, Asp245, Asp333 and cation-π interactions involving 9 basic residues (Lys123, Lys110, etc.), while hydrogen bonds formed with 7 residues (Arg34, Arg38, Asp105, Lys40; 2.9–3.2 Å) (Figure 3B,F). Peptide C interacted with 42 residues: anion-π through Asp333, Asp105, Glu241, Asp245; cation-π through 10 basic residues (Arg34, Arg38, Lys30, etc.); and high-strength hydrogen bonds (minimal distance 2.4 Å) linking Arg34, Arg38, Glu241, Ser141 (Figure 3C,G). Peptide D formed a network across 39 residues, including anion-π interactions with 7 acid residues (Glu23, Asp333, etc.), cation-π interactions with 12 basic residues (Arg125, Arg52, etc.) and a hydrogen bond system comprising Lys40, Arg34, Lys30, Arg125 (Figure 3D,H).

### 3.3. Dynamic Stability of Critical Binding Residues

Asp96, Glu232, and Trp890 emerged as critical mediators of van der Waals interactions throughout the 100 ns MD trajectories with substantial contact maintained to all four peptides despite their divergent binding profiles. Pepetide A maintained stable contacts with Arg30, Ile229, Glu232, Leu293, and Trp890 (Figure 4A). Peptide B primarily affected Tyr29, Arg30, Asp96, Thr98, Glu232, and Trp890 (Figure 4B). Peptide C formed additional stable interactions with Phe573 and Lys806 beyond conserved residues (Figure 4C). Peptide D achieved dynamic stability through Thr130, Lys235, Leu293, and Arg571 (Figure 4D). MM/GBSA analysis of the 90–100 ns equilibrium trajectory quantified the binding free energies for Peptide A, B, C, and D systems as −101.9, −87.72, −129.69 and −141.67 kcal/mol.

Van der Waals interactions dominated these binding events, comprising 86.16–95.51% of total binding energy (Figure 5A–D; Table 2). Peptide D achieved maximal binding affinity (−141.67 kcal/mol) primarily through van der Waals interactions at −131.4 kcal/mol, representing 92.8% of total binding energy. Despite marginal electrostatic contributions (−3.91 to −5.76 kcal/mol), complex stability was maintained by solvation energies ranging from −4.94 to −3.12 kcal/mol. Integrated structural analyses indicated reduced compactness in Peptide A, reflected by an Rg decrease from 3.5 nm to 3.4 nm during 80–100 ns, while Peptide D exhibited minimal Rg fluctuation (ΔRg < 0.05 nm). RMSD analysis showed rapid stabilization of Systems A–C within 30 ns to values of 0.6–0.8 nm. In contrast, Peptide D displayed conformational adjustments, with RMSD at 1.15 nm during the initial 0–20 ns decreasing to 0.9–1.1 nm beyond 50 ns. These findings collectively support hydrophobic stacking as the dominant mechanism for bitter peptide recognition.

The four peptide-TAS2R14 complexes showed distinct structural dynamics during 100 ns MD simulations; Peptide D maintained superior conformational stability via radius of gyration (Rg) fluctuations below 0.05 nm (Figure 5E). For comparison, Peptide A showed reduced compactness, as Rg decreased from 3.5 to 3.4 nm between 80 and 100 ns. Peptides B and C exhibited intermediate perturbations culminating in convergence near 3.5 nm during the final 20 ns. RMSD trajectories revealed distinct binding evolution: Peptides A–C achieved conformational relaxation within 30 ns, stabilizing at 0.6–0.8 nm RMSD before exhibiting minor fluctuations (Figure 5F). Peptide D exhibited two major conformational transitions: extension to 1.15 nm at 20 ns followed by stabilization at 0.9–1.1 nm beyond 50 ns. All complexes retained structural integrity and energetic stability despite increased flexibility, showing RMSF values below 2.0 Å (Figure 5G).

### 3.4. Sensory Evaluation and Validation of the Fluorescence Response

Both the total peptides and the four individual peptides (Peptide A–D) showed moderate bitterness intensity similar to caffeine with bitterness scores mainly ranging from 4 to 6. However, the bitterness of the total peptides was significantly higher than that of any single peptide (Appendix A). The EGFP-TAS2R14 cellular model assessed fluorescence dynamics at incremental concentrations (0.1, 0.2, 0.5, 1 mg/mL) for Peptides A–D and total peptides (Figure 6). Fluorescence intensity exhibited progressive enhancement with peptide concentration. Peptide D elicited maximal response at 1 mg/mL, showing the greatest enhancement versus controls. Peptide C reached near-saturation activation at 0.5 mg/mL, while Peptides A and B demonstrated gradual intensity increases, though their maximal responses remained lower than Peptides C/D. Total peptides showed progressive fluorescence enhancement, achieving significance at 0.2 mg/mL. All peptides induced detectable fluorescence signal enhancement.

### 3.5. Plcb2/Ip_3_ Axis-Mediated Bitter Signal Transduction Mechanism

All treatments significantly elevated *TAS2R14* mRNA levels at 0.5 mg/mL (Figure 7A), consistent with fluorescence data. *TAS2R14* expression in Peptide C groups exceeded caffeine controls at 0.1, 0.2, and 0.5 mg/mL (*p* < 0.01), remaining significant at 1 mg/mL (*p* < 0.05). Expression levels of *GNAT1*, *PLCB2*, and *TRPM5* mRNA exhibited upregulation across increasing concentrations (Figure 7B–D). Peptides A and B significantly enhanced *GNAT1* expression versus caffeine at 0.1–0.5 mg/mL (*p* < 0.01). The *GNAT1* upregulation was restricted to the collective peptide sample at 1 mg/mL (*p* < 0.01), whereas Peptide D exclusively induced significant expression at 0.5 mg/mL (*p* < 0.05). Peptides A and B produced a concentration-dependent upregulation of *PLCB2* (0.2–1 mg/mL; *p <* 0.01). This effect was stronger than that of caffeine in all systems where Peptide D was not present. The collective peptide sample showed elevated *PLCB2* at 0.5 mg/mL (*p* < 0.05), progressing to robust upregulation at 1 mg/mL (*p* < 0.01). *TRPM5* expression analysis revealed that Peptide C induced sustained *TRPM5* upregulation versus caffeine controls from 0.2 mg/mL onward, while Peptide D and total peptides exhibited highly significant *TRPM5* elevation at 1 mg/mL (*p* < 0.01).

The cAMP levels declined with increasing concentrations (Figure 8A), while remaining significantly elevated in Peptides A, B, and D groups relative to caffeine groups (*p* < 0.05). IP3 levels showed different patterns of increase (Figure 8B): Peptide D increased significantly at 0.1 mg/mL (*p* < 0.01); Peptides A and C at 0.2 mg/mL (*p* < 0.01); Peptides B and C at 0.5 mg/mL (*p* < 0.01); and all samples except total peptides increased at 1 mg/mL (*p* < 0.01). These results demonstrate that Peptides A–D effectively activate bitter signaling through significant upregulation of *TAS2R14* and its downstream effectors, simultaneously modulating cAMP/IP_3_ levels, with predominant transduction mediated via the *PLCB2*-*TRPM5*/IP_3_ axis and secondary modulation through the cAMP pathway. Cell viability assays confirmed the absence of cytotoxicity for peptides C/D and total peptides in the range of 0.1–1 mg/mL (Figure 8C), with significant proliferation-promoting effects observed at increasing concentrations. Peptides A and B exhibited mild inhibition at 0.1 mg/mL, while both showed enhanced proliferative potential at higher concentrations (≥0.5 mg/mL). These results indicate the biosafety of bitter peptides in *L. lancifolium* at tested doses and reveal a obvious stimulation of cellular proliferation by Peptides C/D.

### 3.6. Transcriptomic Profiling Reveals Multi-Pathway Modulation of Bitter Signaling

The transcriptomic profile evaluated the regulatory effects of total peptide fractions (P1–P4) and monomeric Peptide C on bitter taste signaling pathways at graded concentrations (0.1, 0.2, 0.5 and 1 mg/mL). KEGG enrichment analysis revealed significant enrichment of DEGs in key pathways: taste transduction, neuroactive ligand-receptor interaction, MAPK signaling, cAMP signaling and endoplasmic reticulum protein processing (Figure 9A). The 0.1 mg/mL total peptide group showed maximal enrichment in calcium signaling pathway, involving 12 differentially expressed genes with seven upregulated. Among these, GRM2 exhibited the highest differential expression with a Log_2_FC value of 2.647 and was directly implicated in downstream bitter signaling. The enrichment analysis of the cAMP pathway identified sixteen differentially expressed genes, nine of which were upregulated. Key genes included FOS and JUN, with Log_2_FC values of −3.17 and −1.89, respectively, compared to the control, while NPPA was upregulated with a Log_2_FC of 3.509. The PI3K-Akt pathway contained the highest number of genes, with eleven upregulated and seventeen downregulated. Meanwhile, the MAPK pathway involved twenty-one differentially expressed genes, comprising seven upregulated and fourteen downregulated. These results suggest that bitter perception is modulated through multiple signaling pathways. This shows that *L. lancifolium*-derived bitter peptides are associated with the activation of key pathways, including calcium signaling, cAMP, and MAPK to regulate downstream transduction. RT-qPCR validation of 7 DEGs (*ATF5*, *PPP2R2A*, *CCNE2, PPP2R5A*, *PDE7A*, *PDE6D*, *PDE8A*) selected from transcriptomic data revealed expression profiles consistent with sequencing trends, demonstrating its accuracy (Figure 9B).

## 4. Discussion

This study provides the first systematic elucidation of bitter peptides in *L. lancifolium* and their molecular mechanism in activating bitter taste signaling through the type 2 bitter taste receptor TAS2R14. In contrast to conventional theories attributing lily bitterness to secondary metabolites [34], this study shifts the focus on bitter peptides through an integrated framework of peptidomics, computational screening, and in vitro functional validation. Our peptidomic analysis revealed a significantly distinct peptide profile in the flesh of bitter-tasting *L. lancifolium* compared to low-bitterness varieties, with upregulated peptides accounting for 46.27% of the total. High-throughput docking of 214 upregulated peptides against 25 bitter taste receptors revealed that TAS2R14 exhibited the strongest binding affinity (avg. binding energy: −119.73 kcal/mol), identifying it as the core receptor for bitter peptides. This finding aligns with functional divergence patterns in bitter receptor families: the high structural plasticity of the extracellular loops of TAS2R14 enables the accommodation of various hydrophobic ligands [35]. As a prominent bitter taste receptor, TAS2R14 is located in both gustatory tissues and extraoral organs, extending its distribution to critical systems such as cardiac muscle and smooth muscle of the airways [36]. Activation of TAS2Rs induces dissociation of the gustducin subunit, releasing Gβγ dimers that inhibit the signaling of the acetylcholine receptor through competitive disruption of Gq coupling, thus suppressing Gq-mediated Ca^2+^ mobilization. This mechanism mediates smooth muscle relaxation of the airways with concomitant suppression of pulmonary inflammation and tissue remodeling, establishing a multi-target therapeutic strategy for chronic obstructive pulmonary disease [37]. Therefore, we hypothesize that bitter peptides in *L. lancifolium* may activate respiratory TAS2Rs to mediate the molecular basis of the traditional lung-moistening and cough-suppressing efficacy. This hypothesis deserves future validation in airway cell or animal models.

Molecular coupling analysis revealed that one interaction mechanism between the four prioritized peptides (A–D) and the bitter receptor TAS2R14 involves synergistic effects mediated by hydrogen-bond networks and ion-π interactions. The basic residues Arg34, Arg38, Lys30, and Lys110 act as hydrogen bond donors, and the acid residue Glu241 serves as a proton acceptor, generating a complementary polar microenvironment. These residues cooperatively maintain a stable charge distribution network in the TAS2R14 binding pocket: the cationic centers of the basic residues engage in ion-π interactions with aromatic rings of the peptides, while the anionic charge of Glu241 achieves electrostatic balance through salt bridges or hydrogen bonds. MD simulations identified Asp96, Glu232, and Trp890 as critical residues driving van der Waals contacts. We propose that Trp890 functions as a hydrophobic core, forming stable hydrophobic stacking with nonpolar side chains of bitter peptides via van der Waals forces. This aligns with established evidence that tryptophan residues play a key role in bitter perception due to their strong hydrophobicity and conserved positioning in taste receptors [20].

Asp96 and Glu232, as negatively charged acid residues, directly anchor the polar groups of ligands through hydrogen bonding. Calculations of binding through hydrogen bonds, van der Waals forces and electrostatic interactions, including potential anion effects π [38,39]. The binding free energy analysis quantified solvation energy reduction (−4.94 to −3.12 kcal/mol), confirming water exclusion as a driver for enhanced van der Waals interactions and consequent complex stabilization [40]. Future functional optimization may involve structural modifications of the peptide targeting the conserved residues Asp96, Glu232, and Trp890. EGFP-HEK293T cell assays confirmed that all four peptides improved fluorescence intensity. Peptide D (1 mg/mL) and Peptide C (0.5 mg/mL) exhibited the strongest activation than controls. This finding aligns with the binding free energy of −141.67 kcal/mol for Peptide D.

These peptides activated TAS2R14 downstream signaling pathways while upregulating the expression of its mRNA and effectors including *GNAT1*, *PLCβ2*, and *TRPM5*. This parallels the established mechanism of the piperine TAS2R14 agonist, which activates enteroendocrine cells by increasing the expression of *PLCβ2* and *TRPM5* [41], and matches the molecular signature of the canonical bitter signal cascade *GNAT1*–*PLCβ2*–*TRPM5* [42]. IP_3_ levels peaked significantly at critical concentrations, particularly 0.1 mg/mL for peptide D and 0.2 mg/mL for peptides A/C, and then declined; however, they persisted above caffeine control levels (*p* < 0.01). These results define the bitter peptide-induced activation profile of signaling pathways: *PLCβ2*-*TRPM5*-mediated IP_3_ production operates as the primary mechanism, while the cAMP pathway provides synergistic auxiliary modulation.

The perception of bitter taste initiates with taste transduction (ko04742) and neuroactive ligand-receptor interactions (ko04080). Our study demonstrates that bitter peptides directly activate these pathways by binding to the TAS2R14 receptor. The cAMP (ko04024) and calcium signaling pathways (ko04020) function as the most immediate intracellular messenger routes [43,44]. Activation of TAS2R14 alters intracellular cAMP levels and triggers calcium release thereby converting chemical bitterness into electrical signals [19]. These signals are transmitted to the brain via sensory neurons leading to the subsequent activation of the PI3K-Akt signaling (ko04151) and MAPK (ko04010) pathways which play critical roles in neuronal excitability synaptic plasticity and signal amplification [45,46]. As a warning signal shaped by evolution bitter taste often indicates potentially toxic substances. The relaxin signaling pathway (ko04926) may modulate relaxation and secretion in the digestive tract [47] while enrichments in ECM-receptor interaction (ko04512) and cell adhesion molecules suggest that bitter perception may influences cellular interactions with the extracellular environment. Activation of protein processing in the endoplasmic reticulum (ko04141) reflects the induction of endoplasmic reticulum stress responses to maintain proteostasis under specific stimuli [48]. Thus, our transcriptomic analysis reveals that bitter taste perception represents not merely a simple gustatory pathway but a bioactive signal capable of triggering a range of complex physiological activities. RT-qPCR validation of 7 DEGs confirmed expression trends that are highly concordant with transcriptomic data, supporting their reliability.

This study identifies four bitter peptide sequences in *L. lancifolium* and their receptor interactions enabling targeted bitterness mitigation strategies including specific enzymatic hydrolysis or natural masking agents like cyclodextrins for competitive receptor inhibition. It also provides novel pathways for functional food development. Given the widespread expression of bitter taste receptors in the gastrointestinal tract and their role in metabolic regulation [48] these bitter peptides with defined receptor activation effects may be developed into functional ingredients for appetite and blood sugar regulation thereby promoting the transformation of *L. lancifolium* into high-value health products. The integrated innovative approach developed in this study, which combines peptidomics, computational screening, and functional validation, efficiently narrows down candidate peptides from thousands of sequences. Subsequent high-throughput molecular docking with relevant taste receptors such as the umami receptor T1R1/T1R3 and bitter taste receptors TAS2Rs offers a rational strategy for prioritization, thereby reducing the time and financial investment. This workflow can be directly applied to identify umami peptides from sources like soybeans and shiitake mushrooms or to investigate bitter taste as a critical quality attribute in medicinal plant resources.

## 5. Conclusions

This study establishes bitter peptides as one of key contributors to the bitterness of *L. lancifolium* through an integrated approach. Peptidomic analysis revealed 8479 unique peptides, with 46.27% upregulated specifically in *L. lancifolium* flesh. Combined computational and cellular assays identified four core bitter peptides that stably bind TAS2R14, activating a downstream signaling pathway dominated by the PLCB2-IP_3_-TRPM5 axis. The identified peptide fraction was biosafe and exhibited cell-proliferative activity at 0.1–1 mg/mL. These findings provide a strategy for bitterness control in food processing, enabling the screening and breeding of low-bitterness lily varieties through peptide and receptor-based markers, and establish a research paradigm for exploring taste-active peptides from other plant sources. However, given that the protein structures of most bitter taste receptors remain unelucidated, this study cannot exclude the potential secondary contribution of other bitter receptors to the perception of lily bitterness. Furthermore, this work supports the targeted development of lily-derived bioactive peptides for functional foods, enhancing the utilization of lilies as a plant-based resource.

## Figures and Tables

**Figure 1 foods-14-04056-f001:**
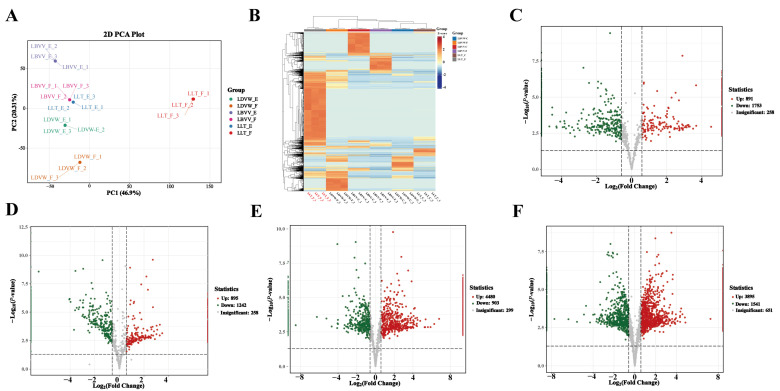
Peptidomics Analysis of Six Samples: PCA, Clustering, Differential Expression, and Functional Enrichment. (**A**). Principal component analysis of six sample groups (LLT_E, LBVV-E, LDVW-E, LLT_F, LBVV-F, LDVW-F). (**B**). Hierarchical clustering heatmap of peptidomic profiles. (**C**–**F**). Volcano plots of differential peptide screening in four comparative groups, respectively, for LLT_E vs. LBVV-E, LLT_E vs. LDVW-E, LLT_F vs. LBVV-F, and LLT_F vs. LDVW-F.

**Figure 2 foods-14-04056-f002:**
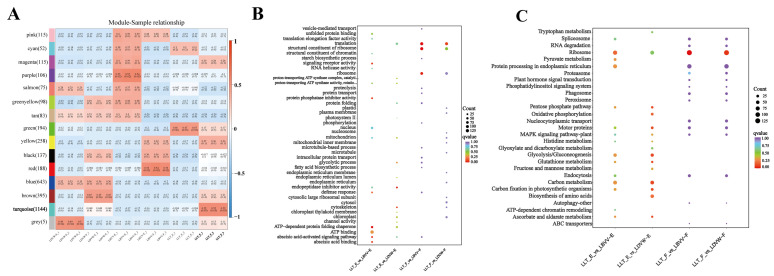
(**A**): Weighted peptide co-expression network analysis (WPCNA) of peptidomic data. (**B**,**C**): GO and KEGG analysis of four differential comparisons.

**Figure 3 foods-14-04056-f003:**
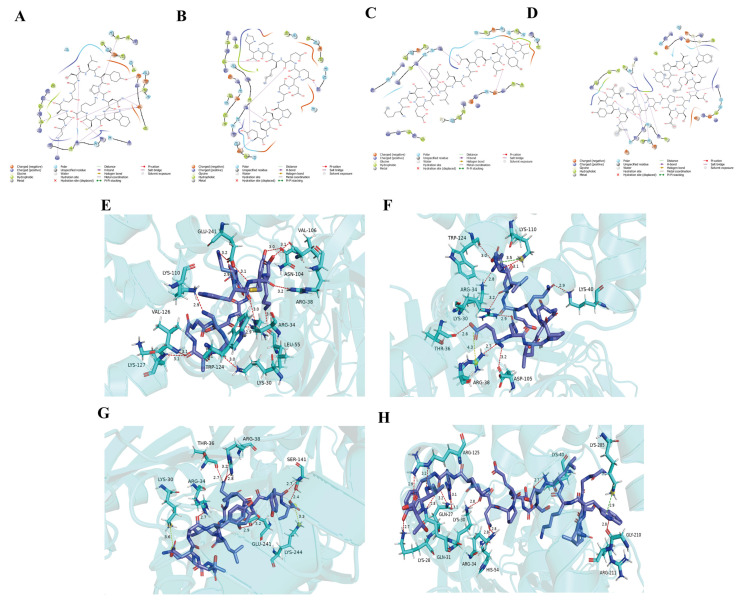
Molecular Docking of Four Peptides (**A**–**D**) with TAS2R14: Interaction Forces, Hydrogen Bonding, and Residue-Specific Dynamics. (**A**–**D**): Key interaction forces and residue contributions in complexes of Peptide (**A**–**D**)-TAS2R14. (**E**–**H**): Hydrogen bond analysis of the (**A**–**D**) peptide-TAS2R14.

**Figure 4 foods-14-04056-f004:**
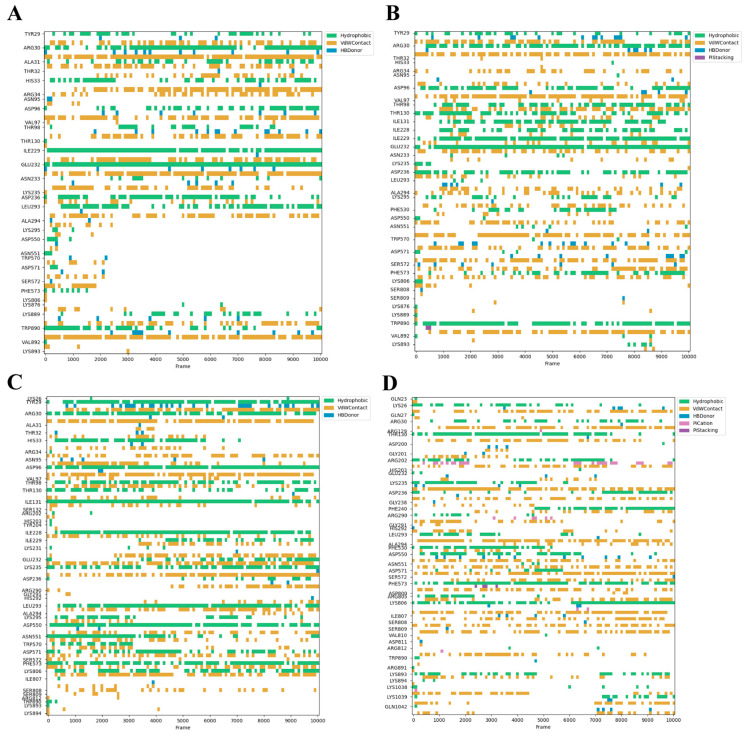
(**A**–**D**): Dynamic interaction of the peptide (**A**–**D**) with critical TAS2R14 residues.

**Figure 5 foods-14-04056-f005:**
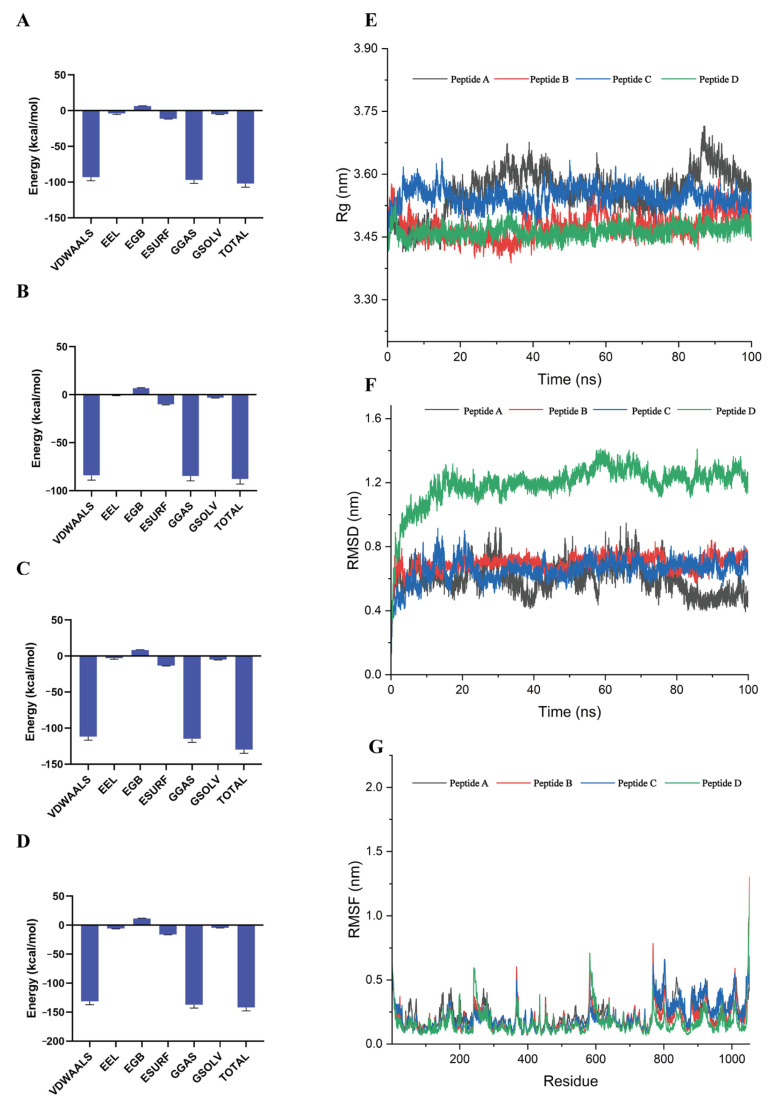
Analysis of Binding Free Energy and Structural Dynamics in Bitter Peptide-TAS2R14 Complexes: MM/GBSA and Conformational Stability Metrics. (**A**–**D**): MM/GBSA Binding Free Energy Profiles of Bitter Peptides (**A**–**D**) in Complex with TAS2R14. (**E**–**G**): Conformational Dynamics of Bitter Peptides (**A**–**D**): Global Structural Compactness, Backbone Stability, and Residue Flexibility.

**Figure 6 foods-14-04056-f006:**
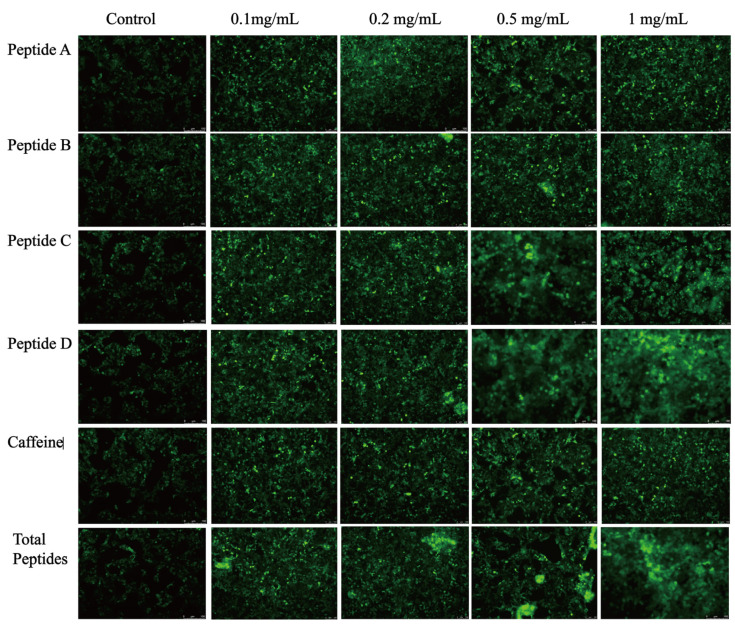
Fluorescent Response of TAS2R14 Receptor Activation by Peptides at Graded Concentrations in a Stable Cell Expression System.

**Figure 7 foods-14-04056-f007:**
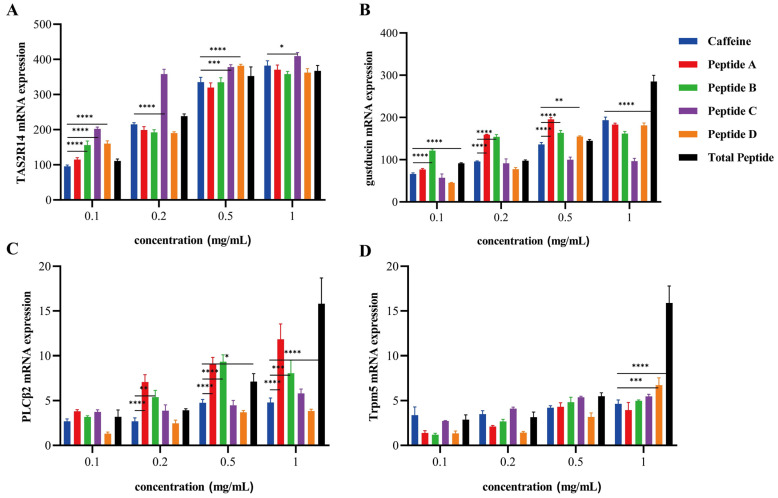
Modulation of Key Bitter Taste Signaling Components by Peptides A–D and Total Peptide. (**A**–**D**): *TAS2R14*, *GNAT1*, *PLCB2*, and *TRPM5* mRNA expression levels. Data are presented as mean ± SEM; * *p* < 0.05, ** *p* < 0.01, *** *p* < 0.001, **** *p* < 0.0001 vs. control group.

**Figure 8 foods-14-04056-f008:**
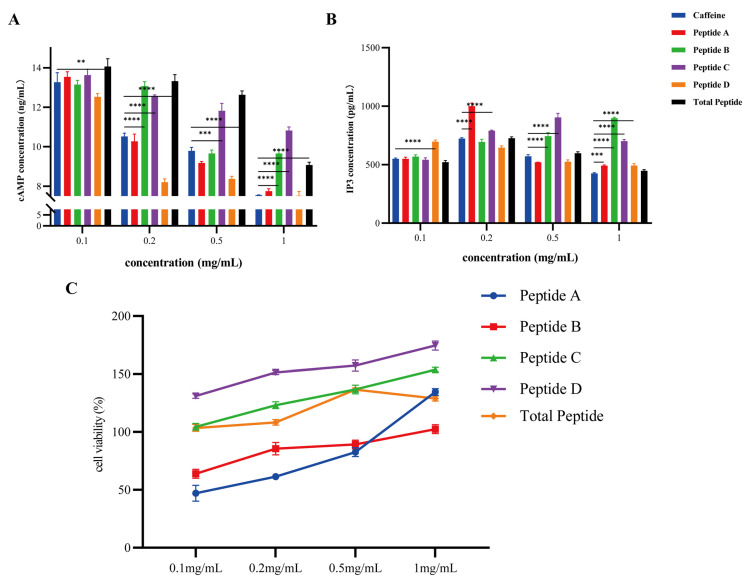
(**A**,**B**): Dynamic changes in cAMP and IP_3_ levels. (**C**). Cell Viability of HEK293T Cells Treated with Peptides ((**A**–**D**) and Total) at different concentrations. Data are presented as mean ± SEM; ** *p* < 0.01, *** *p* < 0.001, **** *p* < 0.0001 vs. control group.

**Figure 9 foods-14-04056-f009:**
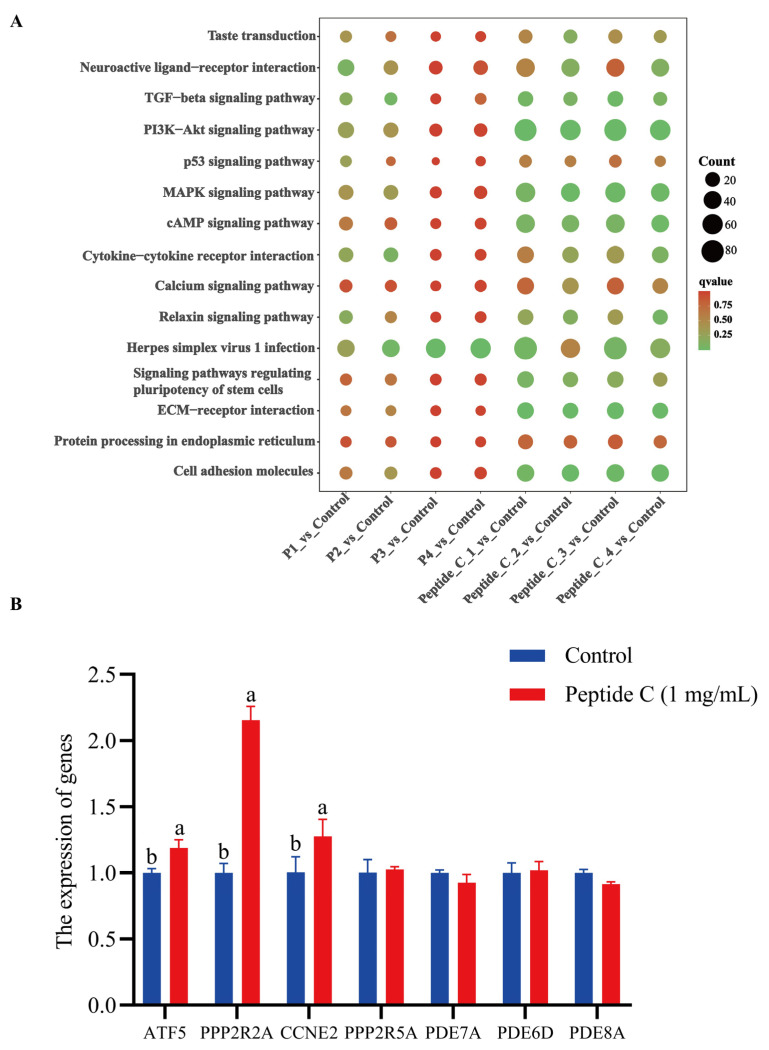
KEGG Pathway Analysis and Differential Gene Expression Verification from Transcriptome Sequencing. (**A**). KEGG Pathway Enrichment Analysis for Differential Comparison Groups. (**B**). Verification of the expression trends of 7 DEGs. (a, b) above the bars indicate statistically significant differences as determined by one-way ANOVA with post-hoc test (*p* < 0.05).

**Table 1 foods-14-04056-t001:** Statistical of Bitter Receptor-Peptide Docking (MS Response Intensity ≥ 10).

Bitter Taste Receptor	Number of Effectively Bound Peptides	CDOCKER Interaction Energy (kcal/mol)
TAS2R3	2	−73.38
TAS2R4	3	−65.18
TAS2R7	2	−49.82
TAS2R10	1	−167.33
TAS2R14	76	−119.73
TAS2R39	1	−139.65
TAS2R41	44	−98.12
TAS2R46	121	−114.88

**Table 2 foods-14-04056-t002:** MM/GBSA binding energy decomposition for the 90–100 ns trajectory.

Energy (kcal/mol)	Peptide A	Peptide B	Peptide C	Peptide D
△Evdw	−93.05	−83.88	−111.74	−131.4
△Ele	−3.91	−0.72	−2.98	−5.76
△Esurf	6.38	6.87	8.28	11.44
△EGB	−11.32	−9.99	−13.25	−15.96
△EGas	−96.96	−84.6	−114.72	−137.16
△Esolv	−4.94	−3.12	−4.97	−4.51
△EBind	−101.9	−87.72	−129.69	−141.67

## Data Availability

The raw sequencing data generated in this study have been deposited in the National Genomics Data Center (NGDC) of the Beijing Institute of Genomics of the Chinese Academy of Sciences, with the following BioProject accession numbers: PRJCA042170 for transcriptomic sequencing.

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
