# Peer review of "Identification of Bitter Peptides in Lilium lancifolium Thunb.; Peptidomics, Computational Simulation and Cellular Functional Assays"

_foods, 2025, doi:10.3390/foods14234056_

Round 1
Reviewer 1 Report
Comments and Suggestions for Authors
This study aligns well with foods by exploring bitterness-related peptides in an edible plant, linking peptidomics with sensory and functional attributes. The integration of computational simulation and cellular assays enhances the understanding of taste-active components, contributing to the improvement of food quality, the development of flavor modulation strategies, and the creation of healthier plant-based food products. However, there are some concerns and suggestions that demand consideration from authors to improve the manuscript.
Title & Abstract
1. The title is informative, but somewhat lengthy. Consider simplifying or restructuring to improve readability while preserving the key concepts. Therefore, I suggested modifying the title in this way: “Identification of Bitter Peptides in Lilium Lancifolium Thunb.; Peptidomics, Computational Simulation and Cellular Functional Assays”
2. The abstract is comprehensive but dense. It would benefit from clearer separation of results and implications to enhance readability.
3. Consider briefly indicating the novelty more explicitly, such as how this work differs from previous studies on lily bitterness.
Introduction
4. The introduction effectively establishes background, but some historical context could be shortened to focus more directly on the current knowledge gap and rationale.
5. Clarify the rationale behind selecting only four peptides for functional validation from the large peptide pool.
6. Additional recent literature on plant-derived bitter peptides in food systems could further support the scientific foundation.
Materials & Methods
7. The workflow sequence (peptidomics → screening → molecular docking → MD → cellular validation → transcriptomics) is complex; a flowchart would help guide readers.
8. The manuscript discusses bitterness in sensory terms, but the human sensory panel design requires clearer justification (training level, reproducibility).
Results & Discussion
9. The discussion successfully links findings to broader taste perception mechanisms; however, consider reducing the repetition of results already described earlier.
10. The potential relevance to food product development could be addressed more prominently to strengthen practical impact.
11. The transcriptomic analysis identifies multiple signaling pathways; however, a deeper discussion is needed to link these pathways to their physiological relevance in food perception.
12. Some figures are visually dense; labeling key residues and interactions in docking diagrams could improve clarity for readers unfamiliar with structural biology.
13. Demonstrating experimental receptor specificity (e.g., comparing TAS2R14 with another TAS2R subtype) would strengthen the mechanistic claims.
14. Consider adding a short paragraph in the discussion on potential practical applications in food breeding, bitterness masking, or functional food development.
Conclusion
15. The conclusion effectively summarizes the key findings, but should be condensed to provide sharper emphasis and highlight the translational value of food industry bitterness control strategies more explicitly.
16. Consider adding a forward-looking statement on how this peptide identification framework can be applied to other edible plant systems.
Author Response
Comments 1: [The title is informative, but somewhat lengthy. Consider simplifying or restructuring to improve readability while preserving the key concepts. Therefore, I suggested modifying the title in this way: “Identification of Bitter Peptides in Lilium Lancifolium Thunb.; Peptidomics, Computational Simulation and Cellular Functional Assays”]
Response to Comments 1:
Thank you for this valuable suggestion. We agree that the original title was somewhat lengthy, and your proposed version is more concise and readable while accurately reflecting the core content and methodologies of our study.
We have accordingly revised the title to: "Identification of Bitter Peptides in Lilium lancifolium Thunb.: Peptidomics, Computational Simulation and Cellular Functional Assays" (Line 2-3)
Comments 2: The abstract is comprehensive but dense. It would benefit from clearer separation of results and implications to enhance readability.
Response to Comments 2:
Thank you very much for your constructive feedback regarding the density of the abstract. We agree that a clearer structural separation between the key findings and their broader implications would enhance readability and impact.
In direct response to your suggestion, we have thoroughly revised the abstract. The main improvements are as follows:
Structural Clarification: We have reorganized the content to create a more logical flow, first presenting the research question and methodological approach, followed by a concise summary of the core experimental results, and concluding with the study's primary significance and implications.
Highlighting Implications: The final sentences now explicitly and distinctly summarize the novel contribution of identifying bitter peptides in L. lancifolium and outline the methodological framework our study provides for future research.
The revised abstract is uupdated in the manuscript (Lines 16-36)
Comments 3: Consider briefly indicating the novelty more explicitly, such as how this work differs from previous studies on lily bitterness.
Response to Comments 3:
We sincerely thank you for this valuable suggestion. In response, we have now explicitly highlighted the novelty of our study in comparison to previous research on lily bitterness. Specifically, the key innovations of this work are twofold:
This study represents the first investigation to shift the focus from secondary metabolites to bitter peptides in the context of lily bitterness. While prior research has predominantly centered on small-molecule bitter compounds, our work identifies and characterizes bitter peptides as a novel contributor to the overall bitterness of lilies.
We have established and applied an innovative research strategy that integrates peptidomics, in silico virtual screening, and in vitro functional validation. This comprehensive framework enables a systematic and efficient discovery of bitter peptides.
These points of novelty have been clearly articulated in the revised manuscript in Lines 508–511 and Lines 593–596, respectively.
Comments 4: The introduction effectively establishes background, but some historical context could be shortened to focus more directly on the current knowledge gap and rationale.
Response to Comments 4:
We sincerely thank you for this insightful suggestion. We fully agree that condensing the historical context will allow for a sharper focus on the current knowledge gap and the primary rationale of our study.
In direct response to this comment, we have shortened the introduction by removing the following specific historical description:
“However, ancient pharmacopeias such as the Ben Cao Tu Jing from the Song Dynasty and the Jiu Huang Ben Cao from the Ming Dynasty document differences in the properties of Juandan lily and Longya lily. Longya lily is recorded as bearing white flowers, featuring elongated scales and tasting sweet, while Juandan lily is described as having flowers with black spots and a bitter flavor [10-11].”
Comments 5: Clarify the rationale behind selecting only four peptides for functional validation from the large peptide pool.
Response to Comments 5:
We thank you for raising this important point regarding the selection of peptides for validation. The rationale for selecting the four peptides from the large initial pool was based on a systematic and multi-step screening strategy designed to prioritize the most promising candidates for initial functional validation, given the practical constraints of de novo peptide synthesis. The selection process was as follows:
From the initial 8,479 peptides identified via peptidomics, we first selected those significantly upregulated (46.27%) in the flesh of Lilium lancifolium (the bitter variety) compared to non-bitter varieties. This led us to hypothesize that a substantial number of bitter peptides are present in Lilium lancifolium. Given the high cost of de novo peptide synthesis, we adopted a targeted strategy for initial functional validation. We performed high-throughput molecular docking of the 214 markedly upregulated peptides with 25 bitter taste receptors. Based on comprehensive docking scores, four top-ranked peptides were chosen for synthesis and subsequent functional analysis.
Comments 6: Additional recent literature on plant-derived bitter peptides in food systems could further support the scientific foundation.
Response to Comments 6:
We sincerely thank you for this valuable suggestion. We agree that incorporating recent literature on plant-derived bitter peptides would strengthen the scientific foundation of our work. In response, we have now added a discussion of relevant recent studies on this topic in the manuscript (Lines 81-83).
Comments 7: The workflow sequence (peptidomics → screening → molecular docking → MD → cellular validation → transcriptomics) is complex; a flowchart would help guide readers.
Response to Comments 7:
Thank you for your insightful comment regarding the complexity of the workflow sequence. We fully agree that a visual summary would greatly enhance readers’ understanding of the study design and experimental logic.
As recommended, we have supplemented a comprehensive flowchart to illustrate the step-by-step workflow of the entire study. This flowchart is now included in the revised manuscript as Figure 1.
Comments 8: The manuscript discusses bitterness in sensory terms, but the human sensory panel design requires clearer justification (training level, reproducibility).
Response to Comments 8:
We sincerely thank you for raising this important point regarding the justification of our human sensory panel design. We agree that a clearer description of the panelists' training and the measures taken to ensure reproducibility is crucial. In direct response to this comment, we have now expanded the relevant section in the manuscript (Lines 118-134) with a detailed description of the panel's composition, training, and evaluation procedure.
The added text is as follows:
A human sensory panel was established to evaluate the bitterness intensity of the synthesized peptides. Eight panelists (aged 18-45 years, including students and staff) were recruited from the laboratory and surrounding communities. All participants were confirmed to be non-smokers, with no known taste disorders or food allergies, and were not taking medication that could affect taste perception. Prior to formal evaluation, all panelists underwent a standardized training session. This session included: An introduction to the sensory evaluation procedure and the specific task of rating bitterness intensity; Familiarization with the reference standards: a 0.05% (w/v) caffeine solution was used as an anchor for "moderate bitterness" and deionized water was used as a "no bitterness" control; Practice evaluations of the reference samples and a series of peptide solutions at different concentrations to ensure consistent understanding and use of the rating scale. During the formal testing, each peptide sample and control was evaluated in triplicate by each panelist on separate days to assess reproducibility. The presentation order of samples was randomized for each session to avoid bias. Panelists were instructed to rinse their mouths thoroughly with deionized water between samples and to wait for a minimum of 1-minute interval to minimize carry-over effects.
Comments 9: The discussion successfully links findings to broader taste perception mechanisms; however, consider reducing the repetition of results already described earlier.
Response to Comments 9:
We sincerely thank you for this insightful suggestion. We agree that minimizing the repetition of results in the Discussion section is crucial for maintaining a concise and impactful narrative. In response, we have carefully reviewed the entire Discussion and have streamlined it by removing or condensing multiple instances where detailed results were unnecessarily reiterated.
Comments 10: The potential relevance to food product development could be addressed more prominently to strengthen practical impact.
Response to Comments 10:
We are grateful for this valuable suggestion to highlight the practical implications of our work. In response, we have significantly strengthened the discussion of potential applications in food product development by adding a dedicated paragraph in the Discussion section (Lines 585-592).
Comments 11: The transcriptomic analysis identifies multiple signaling pathways; however, a deeper discussion is needed to link these pathways to their physiological relevance in food perception.
Response to Comments 11:
We sincerely thank you for this insightful comment. We agree that a more detailed discussion linking the identified signaling pathways to their physiological relevance in food perception.
We now provide a deeper mechanistic interpretation that explicitly connects each key signaling pathway, including taste transduction (ko04742), neuroactive ligand-receptor interaction (ko04080), cAMP (ko04024), calcium signaling (ko04020), PI3K-Akt (ko04151), and MAPK (ko04010), to its specific role in the physiological process of bitter taste perception. This includes detailing the signal progression from initial receptor binding and intracellular second messenger events to neuronal signal transmission and potential broader physiological effects in the gastrointestinal tract.
The revised text are now included in the manuscript (Lines 565-583).
Comments 12: Some figures are visually dense; labeling key residues and interactions in docking diagrams could improve clarity for readers unfamiliar with structural biology.
Response to Comments 12:
We thank you for this valuable suggestion aimed at improving the clarity of our molecular docking figures. In response to this comment, we have undertaken the following revisions to enhance visual accessibility:
The original, visually dense Figure 2 has been split into two separate figures (now Figure 3 and Figure 4) to allow for a clearer and more detailed presentation of the data.
In the new Figure 3 (Panels A-D), the key interacting residues are now explicitly labeled on the diagrams themselves. Furthermore, the corresponding figure captions clearly describe the specific types of interactions (e.g., hydrogen bonds, hydrophobic interactions) formed by these residues.
Additionally, Figure 3 (Panels E-H) are dedicated to visualizing and labeling the specific hydrogen bond distances, providing details for these critical interactions.
Comments 13: Demonstrating experimental receptor specificity (e.g., comparing TAS2R14 with another TAS2R subtype) would strengthen the mechanistic claims.
Response to Comments 13:
We thank you for this insightful comment regarding receptor specificity. We agree that a comparative analysis with other TAS2R subtypes would provide additional mechanistic depth. However, the primary objective of our current study was to establish the fundamental presence and activity of bitter peptides in lily, a question that had not been previously explored, rather than to demonstrate their specificity for TAS2R14.
Our initial virtual screening against a panel of 25 bitter taste receptors revealed that the identified peptides could potentially interact with several TAS2Rs, which is consistent with the known broad and overlapping ligand specificity of this receptor family. We selected TAS2R14 for further experimental validation in a stable cell-based system specifically because it demonstrated, on average, the strongest binding affinity (most negative docking scores) and engaged with a large number of our candidate peptides, positioning it as a highly relevant and major mediator of the perceived bitterness in our system.
Comments 14: Consider adding a short paragraph in the discussion on potential practical applications in food breeding, bitterness masking, or functional food development.
Response to Comments 14:
We sincerely thank you for this excellent suggestion to enhance the practical impact of our discussion. In direct response to this comment, we have now added contents in the Discussion section (Lines 585-592) that explicitly outlines the potential applications of our findings.
Comments 15: The conclusion effectively summarizes the key findings, but should be condensed to provide sharper emphasis and highlight the translational value of food industry bitterness control strategies more explicitly.
Response to Comments 15:
We are deeply grateful for your invaluable suggestion. We agree this guidance and have accordingly revised the Conclusion section to make it more concise, while now more explicitly and prominently highlighting the translational value of our findings for developing bitterness control strategies in the food industry. We sincerely hope that the revised version meets with the reviewer's approval and thank you once again for providing this critical feedback, which has substantially improved the impact of our manuscript.
Comments 16: Consider adding a forward-looking statement on how this peptide identification framework can be applied to other edible plant systems.
Response to Comments 16:
We are grateful for your excellent and forward-looking suggestion. In full accordance with your guidance, we have now incorporated a dedicated forward-looking statement into the Discussion section (Lines 592-600). This addition explicitly outlines how the integrated framework established in our study can be productively applied to identify taste-active peptides in other edible plant systems, such as soybeans and shiitake mushrooms for umami peptides, or other medicinal plants for bitterness profiling. We sincerely hope that this revision meets with your approval and thank you once again for your invaluable insight, which has significantly broadened the impact and applicability of our research.
Reviewer 2 Report
Comments and Suggestions for Authors
This study presents an interesting and well-conducted investigation combining peptidomics, molecular docking, and cellular assays to identify and characterize bitter peptides in Lilium lancifolium. The topic is novel and contributes useful insights into the biochemical basis of bitterness in lilies, moving beyond the traditional focus on secondary metabolites. The experimental design is sound and the results are clearly presented; however, some minor issues should be addressed.
The main weakness lies in the introduction and conclusion sections. The research aim and hypothesis are not stated clearly enough at the beginning. The introduction should explicitly mention the aim of the study. The conclusions currently read more like a summary of results rather than a synthesis of meaning. They should better highlight the overall significance of the findings, acknowledge study limitations and suggest future directions such as potential applications or functional peptide development.
Minor textual corrections are also needed. In line 85, there is a repetition (“This study This study aims to identify”) that should be corrected. In line 278, the word “totol” should be replaced with “total.” The reference format in line 323 is inconsistent and should be standardized. Table S2 lacks units, which need to be added for clarity. In addition, ensure that abbreviations are defined on first use, gene and protein names follow consistent formatting, and figure or table legends are self-explanatory.
Overall, this is a valuable and original contribution that will be suitable for publication after minor revisions.
Author Response
Comments 1: [The research aim and hypothesis are not stated clearly enough at the beginning. The introduction should explicitly mention the aim of the study.]
Response to Comments 1:
Thanks for your valuable comment regarding the need for a clearer statement of the research aim at the beginning of the study. We agree that explicitly stating the study's objective enhances the focus and clarity of the introduction.
Accordingly, we have revised the manuscript to clearly articulate the research aim. The following text has been added to the Introduction section (Lines 55–61):
“During processing or digestion, many plant-derived proteins generate bitter-tasting peptides [12-13], yet the protein and peptide composition of L. lancifolium has received relatively little research attention. This study seeks to determine whether polypeptide components constitute a source of bitterness in L. lancifolium, thereby establishing a theoretical foundation for accurate bitterness assessment and systematic regulation of its edible and medicinal quality.”
Comments 2: [The conclusions currently read more like a summary of results rather than a synthesis of meaning. They should better highlight the overall significance of the findings, acknowledge study limitations and suggest future directions such as potential applications or functional peptide development.]
Response to Comment 2:
We sincerely thank you for this insightful comment. We agree that a strong conclusion should transcend a mere summary and instead synthesize the broader meaning, implications, and limitations of the work. Following your suggestion, we have thoroughly revised the Conclusion section to:
Highlight the overall significance of our integrated approach in establishing bitter peptides as one source of bitterness in L. lancifolium.
Explicitly acknowledge a study limitation regarding the current structural understanding of bitter taste receptors.
Suggest clear future directions and applications, including practical strategies for bitterness control in food processing, a paradigm for researching other plants, and the targeted development of functional ingredients.
The revised Conclusion is presented below for your review (it is fully incorporated into the manuscript):
This study establishes bitter peptides as one of key contributors to the bitterness of L. lancifolium through an integrated approach. Peptidomic analysis revealed 8,479 unique peptides, with 46.27% upregulated specifically in L. lancifolium flesh. Combined computational and cellular assays identified four core bitter peptides that stably bind TAS2R14, activating a downstream signaling pathway dominated by the PLCB2-IP₃-TRPM5 axis. The identified peptide fraction was biosafe and exhibited cell-proliferative activity at 0.1–1 mg/mL. These findings provide a strategy for bitterness control in food processing, enabling the screening and breeding of low-bitterness lily varieties through peptide and receptor-based markers, and establish a research paradigm for exploring taste-active peptides from other plant sources. However, given that the protein structures of most bitter taste receptors remain unelucidated, this study cannot exclude the potential secondary contribution of other bitter receptors to the perception of lily bitterness. Furthermore, this work supports the targeted development of lily-derived bioactive peptides for functional foods, enhancing the utilization of lilies as a plant-based resource.
Comments 3: [In line 85, there is a repetition (“This study This study aims to identify”) that should be corrected.]
Response to Comment 3:
Thank you for pointing out this oversight. The repeated phrase has been corrected in the manuscript (now appearing at Lines 90-91).
Comments 4: [In line 278, the word “totol” should be replaced with “total.”]
Response to Comment 4:
Thanks for your careful reading and for pointing out this typographical error. We apologize for this oversight. The word "totol" has been corrected to "total" in the revised manuscript (Line 308).
Comments 5: [The reference format in line 323 is inconsistent and should be standardized.]
Response to Comment 5:
Thank you for bringing this inconsistency to our attention. The reference format in the mentioned line has now been carefully checked and standardized to ensure it conforms to the required journal style. The correction has been made in the updated manuscript (Line 346).
Comments 6: [Table S2 lacks units, which need to be added for clarity. In addition, ensure that abbreviations are defined on first use, gene and protein names follow consistent formatting, and figure or table legends are self-explanatory.]
Response to Comment 6:
Thanks for your thorough review and valuable suggestions regarding Supplementary Table S2. We have carefully revised the table to address all the points you raised.
We would like to clarify that the primary data derived from RT-qPCR analysis, such as Cycle threshold (Ct) values and the calculated relative expression levels (e.g., 2^-ΔΔCt), are inherently dimensionless and do not have associated units. The Ct value represents the amplification cycle at which the fluorescence signal crosses a defined threshold, and the relative expression is a normalized ratio.
However, in direct response to your comment to ensure utmost clarity, we have thoroughly revised Supplementary Table S2. The specific improvements we have made include. Specifically, we have:
Verified that all abbreviations are defined upon their first use in the table, or the accompanying legend;
Ensured consistent formatting for gene and protein names throughout the table;
The Supplementary Table S2 is updated now. We believe these changes have significantly improved the clarity and professionalism of the table.
Thank you again for your insightful comments.
Reviewer 3 Report
Comments and Suggestions for Authors
Reviewer comments:
The article titled “Identification of Four Bitter Peptides in Lilium Lancifolium Thunb. Based on Peptidomics, Computational Simulation and Cellular Functional Assays” discusses new methodological approaches for the identification of peptides responsible for the bitter taste trait in Lilium sp., which are important for antioxidant applications in food substances. The manuscript is well written with good scientific integrity. Nevertheless, the following recommendations may further enhance the manuscript's quality for its intended readers.
L1-2: It is recommended to use italics in scientific words.
L 38-39: It is always recommended to avoid and/or use the keywords for efficient searchability. It would be better to put two separate keywords.
L 106: It is recommended to mention the equipment producer's name, city, and country for the ease of future reproducibility.
L 111-112: Please mention the approval number for sensory evaluation.
L 120-131: The electronic tongue methodology reference should be added for future reproducibility. Please mention what taste traits are tested through the ET technique? Please also specify the calculation method used for the taste traits.
L 132: Please mention the reference if it was reproduced.
L 144: Please mention the manufacturer, city, and country name of the device for future reproducibility.
L 152: Please mention the name of the drying gas.
L 212-213: Please, at first, use the abbreviations of FBS, PBS, DMSO, and or others, and then use the abbreviated terms in the next sections for clarity for the readers.
L 294-296: ET technique usually detects umami, richness, and sourness taste traits as well, which are important in terms of consumer acceptance and presently have commercial importance. If these points had also been mentioned in the results, it may have improved the readability for the intended readers.
Figure 1,2, 5. It is recommended to use larger images and split them into multiple figures instead of combining multiple images into one figure to improve visibility.
L 570: This section focused on the antitussive function of the Lilly tubers in the present study, which was not mentioned in the abstract or introduction section. It is advised to emphasize this in the other sections.
Author Response
Comments 1: [L1-2: It is recommended to use italics in scientific words.]
Response to Comment 1:
Thank you for this suggestion. We have revised the scientific words in Lines 1-2 to italics, as seen in Lines 1-2 of the revised manuscript titled "Identification of Bitter Peptides in Lilium lancifolium Thunb.; Peptidomics, Computational Simulation and Cellular Functional Assays"
Comments 2: [L 38-39: It is always recommended to avoid and/or use the keywords for efficient searchability. It would be better to put two separate keywords]
Response to Comment 2:
We thank for this helpful suggestion. We have revised the keywords by separating the combined term "Molecular docking and dynamics simulation" into two distinct keywords: "Molecular docking;" and "Dynamics simulation;" to improve searchability and clarity (Line 38).
Comments 3: [L 106: It is recommended to mention the equipment producer's name, city, and country for the ease of future reproducibility]
Response to Comment 3:
The samples were lyophilized followed by cryogenic pulverization using a ball mill apparatus from Changsha Miqi Instrument Equipment Co., Ltd. (Changsha, China) (Lines 110-112).
Comments 4: [L 111-112: Please mention the approval number for sensory evaluation.]
Response to Comment 4:
We appreciate your suggestion to include the ethical approval number. In response, we have added the following statement to the manuscript (Lines 138-143): "The research involving human participants... (Approval No: 2025-194). Informed consent was obtained from all individual participants involved in the study."
Comments 5: [L 120-131: The electronic tongue methodology reference should be added for future reproducibility. Please mention what taste traits are tested through the ET technique? Please also specify the calculation method used for the taste traits.]
Response to Comment 5:
Thank you for your valuable comments and constructive suggestions, which have greatly helped improve the quality of our manuscript. We have carefully addressed each of your concerns as follows:
Response to the request for adding electronic tongue methodology reference.
As suggested, we have supplemented the relevant reference for the electronic tongue methodology in the manuscript (see Reference 24).
Response to the inquiry about tested taste traits
The electronic tongue instrument used in this study is the SA402B model from INSENT Corporation (Japan). We focused on the single taste trait of bitterness and only employed the bitterness sensor for detection and analysis. This information has been clearly stated in the revised manuscript (Lines 144-145).
Response to the request for specifying the calculation method for taste traits
We have clarified the calculation methods for taste traits in the manuscript (Lines 146-150). Specifically, Principal Component Analysis (PCA) was used for initial pattern recognition and sample differentiation of the taste data acquired from the sensor array. For the quantitative analysis of the key taste trait (bitterness intensity), a prediction model based on Artificial Neural Network (ANN) algorithms was established. This model effectively maps sensor signals to taste perceptions, and both PCA and ANN are standard, robust approaches in electronic tongue data analysis for taste trait quantification.
Comments 6: [L 132: Please mention the reference if it was reproduced.]
Response to Comment 6:
We thank you for this comment. As requested, we have now added the relevant references to the sample preparation and protein extraction protocols in the revised manuscript (Lines 163-172). The described methodology was adapted from established procedures in the literature, and the specific citations are as follows:
Reference 25 was cited for the general approach to sample homogenization and protein extraction.
Reference 26 was cited for the protocols related to reduction, alkylation, and protein precipitation.
Reference 27 was cited for the peptide purification and quantification steps.
Comments 7: [L 144: Please mention the manufacturer, city, and country name of the device for future reproducibility.]
Response to Comment 7:
We thank you for pointing this out. As suggested, the manufacturer and country of origin for the mass spectrometry device have now been specified in the revised manuscript as "tmsTOF Pro 2 (Bruker, Germany)" (Line 180).
Comments 8: [L 152: Please mention the name of the drying gas.]
Response to Comment 8:
We thank you for this pertinent suggestion. As requested, the specific name of the drying gas, nitrogen, has now been explicitly stated in the revised manuscript (Line 182).
Comments 9: [L 212-213: Please, at first, use the abbreviations of FBS, PBS, DMSO, and or others, and then use the abbreviated terms in the next sections for clarity for the readers.]
Response to Comment 9:
We sincerely thank you for this valuable suggestion to enhance the clarity of the manuscript. As requested, we have now carefully revised the text to ensure that all key terms are defined by their full name upon first use, followed by their abbreviations in parentheses. These abbreviations are then used consistently throughout the subsequent sections. The revised passage (Lines 242-259) now includes the following definitions: Dulbecco's Modified Eagle Medium - High glucose (DMEM-H); Phosphate-Buffered Saline (PBS); Fetal Bovine Serum (FBS); Polymerase Chain Reaction (PCR); Dimethyl sulfoxide (DMSO); Reverse Transcription quantitative Polymerase Chain Reaction (RT-qPCR); Green Fluorescent Protein (GFP).
Comments 10: [L 294-296: ET technique usually detects umami, richness, and sourness taste traits as well, which are important in terms of consumer acceptance and presently have commercial importance. If these points had also been mentioned in the results, it may have improved the readability for the intended readers.]
Response to Comment 10:
We thank you for this insightful comment and for highlighting the broad capabilities of electronic tongue (ET) systems for multi-trait taste analysis. We agree that umami, richness, and sourness are indeed crucial taste attributes for consumer acceptance.
In our study, the experimental scope was focused on quantifying bitterness. We utilized the INSENT SA402B electronic tongue system (Japan) and configured it to employ only the bitterness sensor for the detection and analysis of this single taste trait.
This detail has been stated in the revised manuscript (Lines 144-145).
Therefore, while we fully acknowledge the reviewer's valid point, the data for other taste traits such as umami, richness, and sourness were not acquired in this particular experimental setup. Our results and discussion are thus intentionally focused on the bitterness attribute. We believe this focused approach was appropriate for the specific aims of our study, but we thank the reviewer for the suggestion, which we will certainly consider for the design of future, more comprehensive taste profiling experiments.
Comments 11: [Figure 1,2, 5. It is recommended to use larger images and split them into multiple figures instead of combining multiple images into one figure to improve visibility.]
Response to Comment 11:
We sincerely thank you for this excellent suggestion. We agree that increasing the size of the figures significantly enhances their clarity and readability. In direct response to this comment, we have thoroughly reorganized the figures as follows: The original Figure 1 has been split into two new, larger figures: Figure 1 and Figure 2; The original Figure 2 has been split into two new, larger figures: Figure 3 and Figure 4; The panels (A, B, C, D) from the original Figure 5 are now presented as a new, standalone Figure 7; The panels (E, F) from the original Figure 5 have been merged with the original Figure 6 to form a new, consolidated Figure 8.
Comments 12: [L 570: This section focused on the antitussive function of the Lilly tubers in the present study, which was not mentioned in the abstract or introduction section. It is advised to emphasize this in the other sections.]
Response to Comment 12:
Thank you very much for your valuable comment. We agree that the antitussive function of the lily tubers should be highlighted not only in the Results section but also in the Abstract and Introduction for better coherence and emphasis.
Accordingly, we have duly emphasized this key point in both sections. Specifically:
A statement has been added to the Abstract (Line 17). Relevant descriptions have been incorporated into the Introduction (Lines 47–51).We believe these revisions help to better frame our study and align the content across sections.
Thank you again for your insightful suggestion.
Round 2
Reviewer 1 Report
Comments and Suggestions for Authors
I have reviewed the updated files that you have sent for the manuscript ID: 3988511. The authors have sufficiently justified the comments that were provided. I suggest/recommend that the paper be accepted.